# Does the COVID-19 Pandemic Change Consumers’ Food Consumption and Willingness-to-Pay? The Case of China

**DOI:** 10.3390/foods10092156

**Published:** 2021-09-12

**Authors:** Wei Yue, Na Liu, Qiujie Zheng, H. Holly Wang

**Affiliations:** 1School of Economics, Guizhou University, Guiyang 550025, China; davidyue2020@126.com (W.Y.); CatherinaL@126.com (N.L.); 2Maine Business School, University of Maine, Orono, ME 04469, USA; qiujie.zheng@maine.edu; 3China Academy for Rural Development, Zhejiang University, Hangzhou 310058, China; 4Department of Agricultural Economics, Purdue University, West Lafayette, IN 47907, USA

**Keywords:** food consumption, pandemic, willingness-to-pay, double-bounded

## Abstract

Since COVID-19 was first detected in China in 2019, governments around the world have imposed strict measures to curb the spread of the coronavirus, which substantially impacted people’s life. Consumers’ food consumption behavior has also changed accordingly with reduced grocery shopping frequency, replaced in-person grocery shopping with online shopping, and increased valuation on food. In this paper, we aim to investigate the change in Chinese consumers’ food consumption and their willingness to pay (WTP) for vegetables and meat, using a dataset with 1206 online samples collected between February and March 2020. Consumers’ WTP for vegetables and meat is estimated using a double-bounded dichotomous contingent valuation design, and factors affecting their WTPs are also investigated. Results show that consumers have a higher WTP for these food products during the pandemic, and their WTP is positively affected by their anticipated duration of the COVID-19, their online shopping shares, their direct exposure to infected patients, their gender, and their income. These results imply that the food industry shall try to develop online market channels as consumers are willing to share the costs, while lower-income consumers may not be able to meet their food needs with prices increased beyond their WTP and thus may call for the government’s support.

## 1. Introduction

Unexpected public crises may cause drastic changes in consumer behaviors. The most recent worldwide public crisis is the COVID-19 pandemic which first broke out in Wuhan, China, in December 2019. To control the epidemic, the Chinese government immediately issued a lockdown order in most cities, which affected the food supply chain, and consumers responded quickly with adjustments to their purchasing behaviors. Many families chose to hoard more food products based on the psychology of panic to reduce the risk of being infected [1,2,3]. Some consumers changed their shopping channels from offline supermarkets and wet markets to online stores [4,5]. As COVID-19 quickly became a world pandemic, most countries imposed similar measures to restrict direct human contact and resulted in the same consumer behavior changes [6,7].

Consumers’ willingness to pay (WTP) is often used to analyze their purchasing intentions [8,9,10,11]. WTP usually refers to the maximum amount of money that a consumer is willing to pay in exchange for a unit of goods or service. It is a consumer’s personal valuation of a specific item, with a strong subjective evaluation component. However, instead of estimating WTP for a whole product or service, studies have been focused on WTP for specific attributes of a market product or non-market service. For example, compared to the ordinary food available in markets, people have estimated WTP premiums for the attributes of non-Genetic Modified, organic, geographically-identified, high-quality, and animal welfare foods [12,13,14,15,16]. Meta-analyses summarizing and comparing these WTP studies are conducted [17,18]. Results from WTP studies can also be used as a market segmentation factor for food consumers to promote sustainability as most of these studied attributes are eco-friendly attributes [19,20,21,22,23,24,25].

Studies investigating WTP for the whole product are rather few, except for new products that are not available in the market, such as biobatteries [26]. This is because when a product is sold on the market, revealed preferences can be observed by the market price and purchasing quantity so that there is no need to solicit consumers’ WTP using stated preference. However, during unexpected public crises, market equilibrium is disrupted abruptly, and the WTP for whole products, especially the necessity for food products in daily life, needs to be solicited to understand consumer behavior to avoid food shortage.

The impact of public crisis events on consumer behavior and WTP has attracted wide attention. Scholars have studied WTP for specific products under public crisis events and have achieved meaningful results. Lee et al. [27] studied consumers’ WTP in terms of taxes and fees for specific mad cow disease tests when there was an outbreak of mad cow disease in Korea. Facing the outbreak of H1N1, consumers also had high WTP for a specific vaccine [28]. Zheng et al. [29] studied the WTP for face masks during COVID-19 and found consumers expect a higher price and are willing to pay more for face masks. While these studies are for specific products or services that directly mitigate the adverse effects of public crises, it is especially worth investigating WTP for the essential food products because their market price may increase beyond the normal range of fluctuation due to excessive demand and supply disruption.

Understanding consumers’ WTP for essential food products such as vegetables and meat can provide important consumer side information and help the government make policies to alleviate food shortages and supply chain ruptures. At the same time, the industry can adjust market strategies to satisfy consumer needs. This article will investigate consumers’ WTP for essential food products, vegetables, and meat, during the first outbreak of COVID-19. We fill the gap in the literature that WTP is for the whole product instead of product attributes after disasters while the product is not directly disaster mitigating but rather life essential—food.

In this article, we conducted a contingent valuation study using a double-bounded dichotomous choice approach to estimate consumers’ WTP for vegetables and meat under the influence of the COVID-19 pandemic. We also analyzed the factors affecting WTP. The objectives of this article are (1) to study whether and to what extent Chinese urban consumers are willing to pay for vegetables and meat beyond regular market prices during the COVID-19 pandemic; (2) to study the role people’s expectations of the epidemic’s duration that affect their WTP for vegetables and meat, and; (3) to explore the impact of the COVID-19 pandemic on consumption perspectives for consumers with different socio-demographic characteristics. As the coronavirus variants are emerging, the world keeps fighting. The battle is not over yet, and the pandemic’s impact on food consumption continues. Thus, our study on this issue is valuable in terms of helping understand consumer’s behavior in the COVID-19 environment.

## 2. Research Methods

The contingent valuation method is widely used in non-market valuation. This method is to establish a hypothetical market similar to the research object. Under the premise of this hypothetical market, the consumer WTP is estimated through data obtained from surveys. The method was firstly used by Davis [30] who conducted an empirical study on the recreational value of forest areas in Maine, USA. In 1979, the U.S. Department of Water Resources successively wrote the contingent valuation as one of the basic methods of resource assessment into regulations [31]. Since the 1970s, it has gradually been used in the benefit evaluation of various public goods and related policies, mainly involving outdoor entertainment, air quality, health risks, water quality, nuclear pollution risks, culture and art, and many other fields. Bennett and larson [32] reviewed these early studies and provided a summary.

With the double-bounded dichotomous choice method, survey participants are asked whether they are willing to pay or accept the bid value of a certain amount for the product. Then, depending on their response, they will be asked if they would be willing to pay a higher or lower amount. Thus, the double-bounded dichotomous choice method can collect more information about WTP [33]. This method is considered more efficient than the previous single-bounded method [34,35]. To a certain extent, it reduces hypothetical bias and strategic bias, and more accurately reflects the respondents’ WTP, and improves the accuracy of the research. Thus, we used the double-bounded dichotomous choice contingent valuation design to estimate consumer’s WTP for vegetables and meat.

In this study, the participants were asked whether he or she was willing to buy vegetables or meat if the market price of the products is raised by *B_O_*. The percentages were randomly selected from five situations, namely 15%, 30%, 60%, 100%, and 150%. When the respondents answered “yes” for the first question, they would be asked another higher bid quote of *B_H_* as the second question, otherwise, they would be provided with another lower bid quote of *B_L_*, where *B_L_* < *B_O_* < *B_H_*. In these five price rise scenarios (see Table 1), *B_O_* = 2*B_L_* = *B_H_*/2, except in the first scenario where *B_L_* = 5% instead, indicating that the price increase was very low.

For *WTP*, the respondent’s answer would have the following four possibilities.
(1)$$T=\{\begin{array}{ccc} 1 & WTP<B_{L}, & \mathrm{The}\;\mathrm{answers}\;\mathrm{are}\;(\mathrm{no},\;\mathrm{no})\\ 2 & B_{L}\leq WTP<B_{O}, & \mathrm{The}\;\mathrm{answers}\;\mathrm{are}\;(\mathrm{no},\;\mathrm{yes})\\ 3 & B_{O}\leq WTP<B_{H}, & \mathrm{The}\;\mathrm{answers}\;\mathrm{are}\;(\mathrm{yes},\;\mathrm{no})\\ 4 & B_{H}\leq WTP, & \mathrm{The}\;\mathrm{answers}\;\mathrm{are}\;(\mathrm{yes},\;\mathrm{yes}) \end{array}$$
where *T* is the observed choice indicator variable that falls into one of the four categories. Assume that the *WTP* of the respondent is linear in parameters.
WTP = xβ + θ(2)
where x is a vector of exogenous variables that affect the WTP, β is the corresponding coefficient vector, θ is the residual term and follows normal distribution θ~N(0, σ^2^). The parameters can be estimated by the ordered Probit model (3) using maximum likelihood estimation.
(3)$$\mathrm{Prob}\left(T=t\right)=\{\begin{array}{c} G\left(\frac{B_{L}-x\beta }{\sigma }\right)\\ G\left(\frac{B_{O}-x\beta }{\sigma }\right)-G\left(\frac{B_{L}-x\beta }{\sigma }\right)\\ G\left(\frac{B_{H}-x\beta }{\sigma }\right)-G\left(\frac{B_{O}-x\beta }{\sigma }\right)\\ 1-G\left(\frac{B_{H}-x\beta }{\sigma }\right) \end{array}\;\;for\;t=\{\begin{array}{c} 1\\ 2\\ 3\\ 4 \end{array}$$

## 3. Survey and Data

From February to March 2020, due to the impact of Covid-19, China was in a state of national lockdown. During this period, we conducted an online survey on the consumers’ WTP for vegetables and meat using a reputable survey company to recruit survey participants from its large national panel. In the end, 1206 surveys were collected in three sample cities of Beijing, Wuhan, and Chongqing. Beijing is the capital city, Wuhan is the city where the coronavirus was first detected and experienced the most turmoil, and Chongqing is a city close to Wuhan which was also hit hard by the coronavirus. The respondents were adults of 18 years and older, and were grocery shoppers.

Fresh vegetables and meat refer to two categories of necessary daily foods in the typical Chinese diet. In 2019, per capita consumption of major foods by Chinese residents was 507.7 kg, of which vegetable consumption accounts for 19.4% and meat accounts for 8.1% [29]. They are the two largest food categories by value. Recent data show that the largest food spending by Chinese consumers was on meat and poultry (27.9%), followed by vegetables (18.8%), fruits (13.6%), dairy (9.9%), and fish (8.4%) in 2005 and are expected to change to 21.9%, 18.5%, 13.6%, 12.5%, and 11.6% by 2025, respectively [36,37]. Table 2 is a report of sample descriptive statistics.

Among the 1206 respondents, the proportion of women was slightly over half, and the average age was 34 years old. The largest education level group was the group with a bachelor’s degree, as high as 68%. The proportions of the other two groups, the group with graduate degrees and the group with high school or lower education, were similar, about 7% and 6%, respectively. The remaining 19% of the respondents had associate degrees. The family size was 3.5 people on average, and there was an average of one child under the age of 18 in each household. The average annual family income was 227,800 yuan (about $35,303). These demographic variable values are in line with similar recent studies for the same Chinese urban consumer food consumptions [1,2,38].

The average number of fresh food shopping trips in every two-month period was 24.3 before the Covid-19, or about every other day, but it dropped to 11.8 during the Covid-19 pandemic, which was quite a significant change. During the Covid-19 pandemic, respondents spent an average of 137.5 yuan per week on fresh food, with a variation of 101.3 yuan and a range from 8.3 to 1250 yuan, showing a significant variation. About 87.4% of the survey participants perceived that the price of fresh food had increased during the Covid-19 pandemic compared to the same period last year, however, 1.4% of the interviewees perceived that the price had dropped and 11.2% perceived the price did not change. About 26% of the respondents perceived that the food supply was as adequate as that of the same period last year, but 10% perceived that the supply was significantly lower. Before the Covid-19 pandemic, the proportion of vegetables that respondents purchased online accounted for about 30% of their households’ vegetable purchases and the proportion of meat purchased online was about the same. However, during the COVID-19 pandemic, the proportion of vegetables and meat purchased online increased to 49.0% and 44.6%, respectively. This was because people tried to avoid shopping in stores or markets to mitigate the risk of being infected.

One-third of the participants were recruited from each of the three cities, i.e., Beijing, Wuhan, and Chongqing (The three cities are located in different regions in China and heterogeneity will be taken into consideration in the model). About 29% of the respondents had relatives or friends working as medical staff or faced risks in other front-line positions. There were 8% of respondents that had family members, relatives, or friends who were infected with Covid-19. Only 9% of respondents moved into the current cities from other places within a year. On average, respondents were quite optimistic and expected the epidemic to end in two months. This was not surprising because the most recent epidemic the Chinese experienced was SARS, which occurred in 2003 and ran for only a few months. Their perceptions of the market and consumption were likely based on this underestimated duration of the pandemic.

## 4. Empirical Results and Discussion

As mentioned earlier, we used two common types of food products to demonstrate the impact of the COVID-19 pandemic on consumers’ WTP—vegetables and meat. Ordered Probit regressions of Equation (3) for vegetables and meat on the many possible influential factors are reported in Table 3, each with two alternative specifications. Regression results in equation format are available in the Appendix A. The estimation was completed in STATA (StataCorp LLC, College Station, TX, USA). Various model specifications were examined with alternative forms of independent variables, and the logarithm of the predicted length of the pandemic was used instead of just the predicted length itself. Note that these are coefficients in Equation (2) of consumers’ WTP measured in percentage of price change. In the following, we report the estimated coefficients that were statistically significant, interpret the results, and discuss the findings.

Several variables were statistically significant in the regression. The variable we were most interested in was consumers’ expectations about the ending time of the COVID-19 pandemic. The coefficient was significantly positive at about 30 and 11 respectively for vegetables and meat. That means consumers were willing to pay a 30% higher price for vegetable products and 11% higher for meat products if the expected duration of the pandemic increased by 100%, or doubled. Vegetables were more essential than meats for Chinese consumers, their WTP was higher for the former. This confirmed our expectation that the more pessimistic views consumers had for the COVID-19 and thinking it would last for a longer time, the more likely they would be willing to pay higher prices on hoarding food. Using the logarithm of the variable, we assumed the marginal effect of *Pred* on WTP was not constant. Since it was generally believed at the beginning that the pandemic would last for two months, which would cause great inconvenience in the food distribution system, consumers raised their willingness to pay for the necessities as a rational decision. This was consistent with other studies documenting consumers’ behavior patterns during the COVID-19 pandemic [2,39,40,41]. Consumers’ behavior patterns during the COVID-19 pandemic can be divided into three stages: reacting, coping, and longer-term adapting [39]. In the first stage, consumers perceive the pandemic as a threat and react by hoarding to restore the loss of control, gain security and comfort, and win the competition over the product scarcity due to supply chain disruptions [42,43,44]. The pandemic has lasted a year and a half by now, which means they would have been willing to pay 66 ((ln500 − ln60) × 30) and 25 percent higher prices for vegetables and meat, respectively. Had people known this pandemic would last an unprecedentedly long period like this, they might have had a different WTP at that time.

At the same time, several variables were also significantly positive. If they had family members, relatives, or close friends infected by COVID-19, a consumer’s WTP would increase from 43% to 53% for the two types of foods in the alternative models. This was consistent with our intuition. When family and friends in their close circle get infected, people can truly feel the threat of the virus and are willing to accept higher prices for goods to save more foods at home in case they are also exposed to the virus. This was similar to other studies which found that consumer behavior is directly linked to the anticipated time spent in self-isolation and the severity of the situation during the COVID-19 pandemic [45].

For the variable of shopping frequency before the outbreak, people who went to the supermarket to buy food more frequently had a higher WTP. Consumers were willing to pay an 0.8% higher price for vegetable products and 0.4% higher price for meat products for each additional shopping trip made during the two months before the epidemic outbreak. This was also reasonable because those who made more shopping trips would habitually have a higher risk of getting infected if they did not alter their shopping behavior, so they had a higher value for food under the new situation of restricted shopping trips.

For demographic variables, compared to the base group of men, women’s WTP ranged from 11% to 14% lower, holding everything else constant. The literature has mixed results about gender heterogeneity on WTP. Many show females tend to have a higher WTP for organic and sustainable food attributes, which is different from our results because women cared more about food safety and quality and food security for their families [2,46,47,48]. On the other hand, there are studies showing women tend to pay less on similar food attributes, consistent with our results [49,50]. This could be a result that they have more experience with the food market and grocery shopping, and thus they are less panicked to offer very high price premiums.

Income was positively significant, which is consistent with both the economic consumption theory as well as most empirical studies such as [51,52,53]. People with higher incomes can afford a higher price. This will leave low-income people at a big disadvantage if food price rises during the pandemic.

Relocate referred to whether the consumer was new to the city and was significant at a 1% level across all models, suggesting that if people just moved to the large city, they would have a higher WTP for vegetables and meat. This was also in line with conventional wisdom. When a person moves to a new city, s/he does not have a deep understanding of the grocery logistics of that city, and s/he has no confidence in whether the logistic chain can cope with a larger impact brought by the COVID-19 pandemic, and s/he is willing to pay more to mitigate risks.

For meat products, the coefficient for the number of children at home was about 13, which was significant at the 5% level. This means that for every additional child in the family, consumers were willing to pay 13% more for meat products. For families with children, parents pay more attention to the COVID-19 situation and are concerned that the food supply may be interrupted resulting in higher prices. The protein in meat products is a necessary ingredient for the growth of children, so parents were willing to pay more for meat. There exist studies showing that consumers with children in families are more willing to pay for high-quality foods [54,55].

It is interesting that education is insignificant. Education usually contributes to WTP on food attributes that are new or scientifically advanced such as environmental-friendly or animal-friendly claims in the U.S. [56] and organic foods in the United Arab Emirates [52], because people with higher education tend to acquire and comprehend new information for these attributes. However, the value of basic food during the crisis did not require consumers to have a higher education background to be aware, and thus consumers’ WTPs did not differ by the educational background in this case.

With the estimated coefficient, the fitted WTP value for each participant could be calculated as in Equation (2) and are reported in Table 4. Note, because our model in (1) through (3) was fitted with the percentage increase of prices, the WTP result is also expressed in terms of price percentage increase.

Table 4 shows that the average WTP for vegetables and meat was 200 and 141 percent higher than prices in normal times, respectively. This was higher than reported by Wang et al. [3] who estimated that Chinese consumers on average were willing to pay about 60.5% premium for fresh products reserves during the Covid-19 pandemic. Meixner found that consumers were inclined to be willing to pay a higher premium for ensured beef during the COVID-19 pandemic [57]. This means in general, during the COVID-19 pandemic, consumers are willing to pay a much higher price than in regular times. For both meat and vegetable, about 45% of the sample respondents have a WTP above the average value, while all are willing to buy meat and vegetables at a price exceeding the normal price to ensure the supply of themselves and their families.

When analyzing the WTP for meat and vegetables together, we found a positive correlation between the two types of foods. That is, consumers who had a greater (or smaller) WTP for vegetables also had higher (lower) WTP for meat. This was consistent with our intuition. For a rational consumer, the decision he makes comes from his perception of changes in the external environment and available options. In many cases, consumers are more likely to choose an easy and quickly attainable option than an option that was more distant but more valuable [58]. Under the Covid-19 health crisis, consumers face a high degree of uncertainty and are subject to travel restriction measures. To avoid the situation that the household may run out of foods and cannot take shopping trips to purchase them, consumers chose to pay premiums to stock up their refrigerator and freezer. In addition, consumers may experience value conflicts while making food choices and will adopt logical and feasible strategies to achieve a balanced state [59]. Goals are critical for determining value and affecting consumers’ choices [60]. When they feel that the COVID-19 pandemic is serious and may take a long time to recover, they increase their WTP for both products at the same time.

## 5. Conclusions

Covid-19 is a global issue that deserves continuous attention in the coming years. The pandemic disrupted the normal order of markets, trade, and supply chains in various countries, and affected all aspects of people’s life. One of the greatest impacts on consumers is to increase the uncertainty of their food accessibility due to possible food supply interruptions.

Through the study, we find that the epidemic has a significant impact on consumers’ expenditure on fresh foods as most of them reported perceiving food prices increase. During the Covid-19 pandemic, the city lockdown has led to an asymmetry of information about food prices and supplies. The government should ensure the transparency of food market information for citizens, which can help citizens stock up food rationally and ease their panic.

On average, consumers are willing to pay higher prices for food, because they feel the pressure of a possible food supply interruption and are willing to pay more to guarantee sufficient foods for the family.

Consumers’ concerns about the increased cost of the food supply chain as well as their panic about the future have significantly increased their willingness to pay for typical food products. The higher-income consumers are willing to pay more than the lower-income ones. This is consistent with the consumption theory. This means the low-income people may not be able to obtain adequate foods if prices rise beyond their willingness to pay levels. This should warrant the public and the government’s attention. During a crisis like the COVID-19 pandemic, the lower-income population may have a tighter budget and a higher financial need, which calls for the consideration of government relief plans.

The positive effect of the pandemic duration expectation on the willingness to pay together with Chinese consumers’ over-optimistic estimation of 60 days suggests that consumers were not prepared for a prolonged pandemic like this one. The results show that the longer consumers believed the Covid-19 pandemic would last, the more likely they are willing to pay higher prices on hoarding vegetables and meat. We draw implications with a caveat since at this point the pandemic has lasted for about one and half years, much longer than the average duration Chinese consumers expected, i.e., about 60 days, at the time they completed the survey. The sudden outbreak of the pandemic changed consumers’ behaviors in terms of reducing shopping trips, hoarding foods, and paying higher prices for essential food products. However, with the prolonged pandemic situation, consumers have experienced fatigue which also impacts their consumption behaviors. Our results may not have sufficient prediction power for this long duration of the pandemic. This is the major limitation of our research and further research examining consumer behaviors and choices over a longer duration of the COVID-19 pandemic may fill this gap. This becomes possible and necessary as the COVID-19 pandemic has run near two years, longer than most people’s expectations.

Consumers take measures themselves to cope with the crisis by reducing their shopping trips and switching to online shopping. Online shopping itself also induces consumers to pay more to cover the cost of delivery services for the food products. This suggests that the food retail industry should consider selling products on online platforms and be prepared for the cannibalization of traditional offline sales. As consumers’ preferences for online shopping are different than in traditional offline markets [61,62,63], the industry needs to adjust its marketing emphasis. It also gives an opportunity for the logistic industry.

## Figures and Tables

**Table 1 foods-10-02156-t001:** Double Bounded Choice Price Scenarios.

Scenario	*B_L_*	*B_O_*	*B_H_*
1	5%	15%	30%
2	15%	30%	60%
3	30%	60%	120%
4	50%	100%	200%
5	75%	150%	300%

Note: *B_O_* is the price rise level offered in the first question, and *B_L_* and *B_H_* are low and high price rise levels offered in the follow up question.

**Table 2 foods-10-02156-t002:** Descriptive Statistics of Demographics and Perception Variables.

Variable	Description	Mean	Std. Dev.	Min	Max
Female	=1 if female; =0 otherwise	0.56 *	0.49	0	1
Age	Age, in years	34.09	8.19	19	74
Eduhs	=1 if high school or lower education; =0 otherwise	0.06	0.23	0	1
Eduass	=1 if with an associate degree/some college; =0 otherwise	0.19	0.39	0	1
Eduba	=1 if having a bachelor’s degree; =0 otherwise	0.68	0.46	0	1
Hhnum	Number of people in the household	3.53	0.98	1	11
Children	number of children	0.89	0.62	0	3
foodfreq_b	Number of purchases of fresh food before Covid-19 over a two-month period	24.34	14.78	1	60
foodfreq_d	Number of purchases of fresh food during Covid-19 over a two-month period	11.77	8.36	1	60
foodexp_a	Per capita weekly expenditure of fresh food during Covid-19 (yuan)	137.48	101.32	8.33	1250
Foodpriceup	=1 perceive the price of food increases; =0 otherwise	0.87	0.33	0	1
foodprice down	=1 perceive the price of food decreases; =0 otherwise	0.01	0.11	0	1
foodsupplyc	=1 perceive sufficient food supply; =0 otherwise	0.26	0.43	0	1
foodsupplyb	=1 perceive significantly short of food supply; =0 otherwise	0.10	0.30	0	1
Relocate	=1 relocate to the city within a year; =0 otherwise	0.09	0.29	0	1
Pred	Anticipating Covid-19 duration (days)	60.13	36.71	7	150
Ffinfect	=1 if family or close friends infected with Covid-19; =0 otherwise	0.08	0.28	0	1
Ffmed	=1 if family or friends have health care workers or other frontline positions at risk; =0 otherwise	0.29	0.45	0	1
Hhinc	annual household income(10,000 ten thousand)	22.78	12.87	2.5	60
Beijing	=1 if from Beijing; =0 otherwise	0.33	0.47	0	1
Wuhan	=1 if from Wuhan; =0 otherwise	0.33	0.47	0	1
Vegonline_b	online vegetable purchase as a percentage of all vegetable purchases before Covid-19	30.65	23.86	0	100
Vegonline_d	online vegetable purchase as a percentage of all vegetable purchases during Covid-19	48.96	30.49	0	100
Meatonline_b	online meat purchases as a percentage of all meat purchases before Covid-19	29.74	24.38	0	100
Meatonline_d	online meat purchases as a percentage of all meat purchases during Covid-19	44.60	30.82	0	100

* The mean of a dummy variable taking values of 0 and 1 only represents the share of the observations with value 1, and a share of those taking value 0 is then 1 minus the mean value.

**Table 3 foods-10-02156-t003:** Regression Results.

Variable	Vegetable		Meat	
	Model 1	Model 2	Model 1	Model 2
Ln(Pred)	30.06 ***	30.56 ***	11.35 ***	10.77 **
	(5.06)	(5.08)	(4.21)	(4.24)
Ffinfect	44.56 ***	49.77 ***	43.22 ***	52.57 ***
	(14.15)	(14.05)	(11.84)	(11.81)
foodfreq_b	0.83 ***	0.81 ***	0.43 **	0.55 ***
	(0.25)	(0.25)	(0.21)	(0.21)
foodfreq_d	−0.33	−0.28	−0.11	−0.18
	(1.05)	(1.04)	(0.90)	(0.89)
Female	−13.86 **	−13.35 *	−10.92 *	−10.32 *
	(6.96)	(7.02)	(5.79)	(5.86)
Hhinc	0.56 *	0.53	0.61 **	0.61 **
	(0.33)	(0.34)	(0.28)	(0.28)
Relocate	44.37 ***	48.20 ***	27.32 ***	33.46 ***
	(12.67)	(12.52)	(10.45)	(10.28)
Children	9.99	9.74	12.87 **	13.58 **
	(6.46)	(6.50)	(5.43)	(5.48)
Eduhs	−9.80	−14.45	−8.19	−14.50
	(20.65)	(20.68)	(17.13)	(17.29)
Eduass	−9.16	−13.38	−12.89	−16.10
	(16.68)	(16.61)	(13.92)	(13.99)
Eduba	−9.95	−11.60	−15.89	−17.24
	(14.50)	(14.52)	(12.10)	(12.22)
Ffmed	−4.97	−2.93	7.10	10.14
	(8.09)	(8.00)	(6.72)	(6.70)
foodexp_a	0.20 ***	0.210 ***	0.17 ***	0.19 ***
	(0.04)	(0.04)	(0.03)	(0.03)
online_b	0.47 ***		0.30 **	
	(0.16)		(0.13)	
online_d	0.12		0.42 ***	
	(0.12)		(0.10)	
foodprice_up	−16.73	−12.94	8.54	14.84
	(11.37)	(11.31)	(9.24)	(9.21)
foodprice_down	−20.97	−6.56	1.41	9.21
	(31.46)	(31.41)	(26.08)	(26.09)
foodsupplyc	−7.95		−5.42	
	(8.31)		(6.95)	
foodsupplyb	18.49		17.86 *	
	(12.08)		(10.05)	
Age	0.14	0.02	−0.59	−0.70 *
	(0.46)	(0.46)	(0.38)	(0.38)
Hhnum	0.90	0.49	2.89	2.65
	(4.24)	(4.26)	(3.53)	(3.56)
Beijing	20.62 **	23.79 ***	12.71 *	17.69 **
	(9.08)	(9.07)	(7.56)	(7.58)
Wuhan	8.23	9.16	−5.65	−3.38
	(9.06)	(8.92)	(7.55)	(7.48)
Constant	−79.95 **	−60.64 *	−29.39	−8.518
	(37.07)	(35.67)	(30.29)	(29.66)
Σ	103.1 ***	104.1 ***	87.68 ***	88.97 ***
	(3.30)	(3.33)	(2.69)	(2.74)
Log likelihood	−1699.31	−1706.64	−1776.57	−1791.34

* *p* < 0.1, ** *p* < 0.05, *** *p* < 0.01.

**Table 4 foods-10-02156-t004:** WTP for Vegetables and Meat.

Variable	Mean	Std.Dev	Min	Max
Vegetable	200.42	44.89	0	404.97
Meat	141.48	39.63	39.71	326.04

## Data Availability

Data used in the analysis are collected from a survey conducted by the authors.

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
