# Peer review of "Does the COVID-19 Pandemic Change Consumers’ Food Consumption and Willingness-to-Pay? The Case of China"

_foods, 2021, doi:10.3390/foods10092156_

Round 1

Reviewer 1 Report

The paper deals with an important issue of change consumers’ food consumption and willingness-to-pay during pandemic COVID-19 in China. 

The strengths of the paper are as follows:
- research material from a online survey (N=1206)
- the ability to catch the aspect of willingness-to-pay for food (i.e., vegetables and meat). 

The content of the manuscript shows that the aim is specified. I propose to make some corrections in the material and method section, to complete the Survey and Data and empirical results, check the conclusions and to add Limitations and Future Research Directions.

The paper deals with an important issue of change consumers’ food consumption and willingness-to-pay during pandemic COVID-19 in China. 

The strengths of the paper are as follows:
- research material from a online survey (N=1206)
- the ability to catch the aspect of willingness-to-pay for food (i.e., vegetables and meat). 

The content of the manuscript shows that the aim is specified. I propose to make some corrections in the material and method section, to complete the Survey and Data and empirical results, check the conclusions and to add limitations and future research directions.

Author Response

We thank the reviewer for the constructive comments. In the following, we use bold type face to respond to each of the comment.

The paper is interesting and well written. There are typos in the text, so I suggest the authors to carefully reread what is written (page 8 line 222, consuemers; page 9, line 256, gurantee).

Response: Thank you for your comments. We corrected those typos and also read through the paper and made further editing.

The discussion of results is totally missing. It would be relevant if Authors make a comparison with previous and parallel studies on this argument, also making comparisons between China and other countries.

Response: We have combined the results and discussion section together, as is allowed by the journal format (“ This section may be combined with Results.”).  This is because the comparison of our results and those in existing literature is better carried out for each of the factors affecting WTP.  We added comparisons to parallel studies about China and other countries throughout the Empirical results section [32,33,…, 46].  There is a great deal of similarity between China and other countries. See track changes.  We also changed the subtitle to Empirical Results and Discussion.

Moreover, in the conclusion section more theoretical and practical implications should be added.

Response: We added more details about implications from our results in the Conclusion section.

Reviewer 2 Report

The paper is interesting and well written. There are typos in the text, so I suggest the authors to carefully reread what is written (page 8 line 222, consuemers; page 9, line 256, gurantee). The discussion of results is totally missing. It would be relevant if Authors make a comparison with previous and parallel studies on this argument, also making comparisons between China and other countries. Moreover, in the conclusion section more theoretical and practical implications should be added.

Author Response

We thank the reviewer for the constructive comments. In the following, we use bold type face to respond to each of the comment.  We have made major revision according to these suggestions and the paper is improved a lot. 

The most serious objection is the adoption of a linear form of the regression function for WTP measurement. For some explanatory variables, this approach leads to illogical conclusions. For example, for the variable 'Anticipating COVID end', extending the duration of the pandemic by one year (and in practice it has already lasted almost two years) the WTP for consumers would increase by over 250%! in the case of vegetables and over 70%! for meat, which is unrealistic. In this type of analysis (food demand, food expenditure, etc.) a different form of function is generally adopted, for example, power, logarithmic. I propose to carry out an analysis with such an adjustment.

Response: Thank you very much for this suggestion.  We have tried quadratic and logarithmic forms and selected the logarithmic form.  The results are much more reasonable, especially for the prolonged pandemic.  Results are updated. As we rerun the estimate we found two small data coding errors in the previous version, one is the opposite of male and female, and the other is for one variable missed a few lines.  They are fixed in the new regression. We apologize, but the main results and stories remain the same.  We have updated the result discussions accordingly.

Secondly, the analysis considers only two groups of food expenditure - vegetables and meat. The authors emphasise that these are the two most important groups of expenditure. But they do not indicate specific data to confirm this information, e.g. what share of total food expenditure among the population in China is represented by vegetables and meat. This needs to be supplemented.

Response: We added detailed statistics to the (now) second paragraph in the Survey and Data section.

Thirdly, the research involves respondents from three selected cities. The question arises as to how representative this group is and, therefore, can the conclusions of the analysis be extended to the entire population? Representativeness of the sample is necessary if we want to formulate universal postulates. Therefore, it is advisable to present socio-demographic data (age, education, income, food expenditure, etc.) not only for the surveyed sample, but also averages for the entire population.   

Response: We agree with the reviewer that representativeness of the sample is important and needs to be clearly described. The sample represents urban shoppers in topline cities instead of whole Chinese population, which is now explicitly stated in the objectives in the Introduction section. This is because grocery shopping in supermarkets or online are only relevant in cities and especially large cities, and the topline cities are market leaders. We added details about these three cities in the first paragraph of the Survey and Data section. Also, heterogeneity across the cities is important. We added a footnote to the last paragraph of the Survey and Data section indicating that the three cities are located in different regions in China and we will take into consideration of the location heterogeneity in the model.

In addition, we compared the major socio-demographics in our sample with those in similar studies and summarize the comparison in the following table.

Variables

Hao et al. 2020

Wang et al. 2020

Shi et al. 2020

This paper

Male

0.5

0.55

0.38

0.44

Age

31.6

34.3% > 35

3% ≤18; 74% 18-25; 8% 25-40; 12% 40-50; 4% >50

34

Annual Household Income (Thousand Yuan)

167

29% < 48; 37.1% 48-96; 33.9% > 96

43% ≤ 50; 23% 50-100; 23% 100-200; 8% 200-500; 3% > 500

227.8

Family size

3.33

NA

3.85

3.53

Education years

16

78.2% > 12 years

15.18

15.7

Provinces/Cities

Beijing Shanghai, Guangzhou,

Beijing, Shanghai, Hubei, Guangdong, Zhejiang, Jiangsu, Hebei, and Shanxi

56% in urban China; 44% in rural China

Beijing, Wuhan, Chongqing

Hao, N.; Wang, H. H.; Zhou, Q. The impact of online grocery shopping on stockpile behavior in Covid-19. China Agricultural Economic Review 2020, 12, 459-470.

Wang, E., An, N., Gao, Z., Kiprop, E., & Geng, X. (2020). Consumer food stockpiling behavior and willingness to pay for food reserves in COVID-19. Food Security, 12(4), 739-747.

Shi, M. I. N., XIANG, C., & ZHANG, X. H. (2020). Impacts of the COVID-19 pandemic on consumers' food safety knowledge and behavior in China. Journal of Integrative Agriculture, 19(12), 2926-2936.

These references are added to the data discussion (line 158-159 in new version).  We didn’t include the above table in the text, but are including it to this response.

Finally, the analysis covers two months of the pandemic - February and March 2020 - and it has been ongoing for almost 2 years. It is worthwhile to refer to this situation in the 'Conclusions' section and indicate whether the prolonged crisis situation is influencing changes in buyers' behaviour in terms of their willingness to pay.   

Response: Thanks for this important point. We added explanations about the caveat in our implications in the Conclusion section.

From specific comments:
in objective 3 (line 78-79), replace with socio-demigraphic characters;

Response: Corrected. We have edited the whole paper again.

line 257-258 - how do the authors know that consumers 'understand the food 257 supply cost increase and are willing to share that cost'? It was not the scope of the analysis, then it must be proved.

Response: Thanks. This claim is not well supported with evidence in the paper. We have this claim removed.

Reviewer 3 Report

The article is another in a series of papers on the effects of the COVID pandemic on buyer behaviour in the food market. I believe that the topic is not yet exhausted, so it is good that next study has been produced. The structure of the publication is correct, the objectives are clearly stated, the research methods are carefully described. The conclusions of the analysis are accurate and fulfil the main objectives of the work. Nevertheless, the article needs some improvement.

The most serious objection is the adoption of a linear form of the regression function for WTP measurement. For some explanatory variables, this approach leads to illogical conclusions. For example, for the variable 'Anticipating COVID end', extending the duration of the pandemic by one year (and in practice it has already lasted almost two years) the WTP for consumers would increase by over 250%! in the case of vegetables and over 70%! for meat, which is unrealistic. In this type of analysis (food demand, food expenditure, etc.) a different form of function is generally adopted, for example, power, logarithmic. I propose to carry out an analysis with such an adjustment.

Secondly, the analysis considers only two groups of food expenditure - vegetables and meat. The authors emphasise that these are the two most important groups of expenditure. But they do not indicate specific data to confirm this information, e.g. what share of total food expenditure among the population in China is represented by vegetables and meat. This needs to be supplemented. 

Thirdly, the research involves respondents from three selected cities. The question arises as to how representative this group is and, therefore, can the conclusions of the analysis be extended to the entire population? Representativeness of the sample is necessary if we want to formulate universal postulates. Therefore, it is advisable to present socio-demographic data (age, education, income, food expenditure, etc.) not only for the surveyed sample, but also averages for the entire population.   

Finally, the analysis covers two months of the pandemic - February and March 2020 - and it has been ongoing for almost 2 years. It is worthwhile to refer to this situation in the 'Conclusions' section and indicate whether the prolonged crisis situation is influencing changes in buyers' behaviour in terms of their willingness to pay.   

From specific comments:

in objective 3 (line 78-79), replace with socio-demigraphic characters;

line 257-258 - how do the authors know that consumers 'understand the food 257 supply cost increase and are willing to share that cost'? It was not the scope of the analysis, then it must be proved.

Author Response

We thank the reviewer for the constructive comments. In the following, we use bold type face to respond to each of the comment.

The most serious objection is the adoption of a linear form of the regression function for WTP measurement. For some explanatory variables, this approach leads to illogical conclusions. For example, for the variable 'Anticipating COVID end', extending the duration of the pandemic by one year (and in practice it has already lasted almost two years) the WTP for consumers would increase by over 250%! in the case of vegetables and over 70%! for meat, which is unrealistic. In this type of analysis (food demand, food expenditure, etc.) a different form of function is generally adopted, for example, power, logarithmic. I propose to carry out an analysis with such an adjustment.

Response: Thank you very much for this suggestion.  We have tried quadratic and logarithmic forms and selected the logarithmic form.  The results are much more reasonable, especially for the prolonged pandemic.  Results are updated. As we rerun the estimate we found two small data coding errors in the previous version, one is the opposite of male and female, and the other is for one variable missed a few lines.  They are fixed in the new regression. We apologize, but the main results and stories remain the same.  We have updated the result discussions accordingly.

Secondly, the analysis considers only two groups of food expenditure - vegetables and meat. The authors emphasise that these are the two most important groups of expenditure. But they do not indicate specific data to confirm this information, e.g. what share of total food expenditure among the population in China is represented by vegetables and meat. This needs to be supplemented.

Response: We added detailed statistics to the (now) second paragraph in the Survey and Data section.

Thirdly, the research involves respondents from three selected cities. The question arises as to how representative this group is and, therefore, can the conclusions of the analysis be extended to the entire population? Representativeness of the sample is necessary if we want to formulate universal postulates. Therefore, it is advisable to present socio-demographic data (age, education, income, food expenditure, etc.) not only for the surveyed sample, but also averages for the entire population.   

Response: We agree with the reviewer that representativeness of the sample is important and needs to be clearly described. The sample represents urban shoppers in topline cities instead of whole Chinese population, which is now explicitly stated in the objectives in the Introduction section. This is because grocery shopping in supermarkets or online are only relevant in cities and especially large cities, and the topline cities are market leaders. We added details about these three cities in the first paragraph of the Survey and Data section. Also, heterogeneity across the cities is important. We added a footnote to the last paragraph of the Survey and Data section indicating that the three cities are located in different regions in China and we will take into consideration of the location heterogeneity in the model.

In addition, we compared the major socio-demographics in our sample with those in similar studies and summarize the comparison in the following table.

Variables

Hao et al. 2020

Wang et al. 2020

Shi et al. 2020

This paper

Male

0.5

0.55

0.38

0.44

Age

31.6

34.3% > 35

3% ≤18; 74% 18-25; 8% 25-40; 12% 40-50; 4% >50

34

Annual Household Income (Thousand Yuan)

167

29% < 48; 37.1% 48-96; 33.9% > 96

43% ≤ 50; 23% 50-100; 23% 100-200; 8% 200-500; 3% > 500

227.8

Family size

3.33

NA

3.85

3.53

Education years

16

78.2% > 12 years

15.18

15.7

Provinces/Cities

Beijing Shanghai, Guangzhou,

Beijing, Shanghai, Hubei, Guangdong, Zhejiang, Jiangsu, Hebei, and Shanxi

56% in urban China; 44% in rural China

Beijing, Wuhan, Chongqing

Hao, N.; Wang, H. H.; Zhou, Q. The impact of online grocery shopping on stockpile behavior in Covid-19. China Agricultural Economic Review 2020, 12, 459-470.

Wang, E., An, N., Gao, Z., Kiprop, E., & Geng, X. (2020). Consumer food stockpiling behavior and willingness to pay for food reserves in COVID-19. Food Security, 12(4), 739-747.

Shi, M. I. N., XIANG, C., & ZHANG, X. H. (2020). Impacts of the COVID-19 pandemic on consumers' food safety knowledge and behavior in China. Journal of Integrative Agriculture, 19(12), 2926-2936.

These references are added to the data discussion (line 158-159 in new version).  We didn’t include the above table in the text, but are including it to this response.

Finally, the analysis covers two months of the pandemic - February and March 2020 - and it has been ongoing for almost 2 years. It is worthwhile to refer to this situation in the 'Conclusions' section and indicate whether the prolonged crisis situation is influencing changes in buyers' behaviour in terms of their willingness to pay.   

Response: Thanks for this important point. We added explanations about the caveat in our implications in the Conclusion section.

From specific comments:
in objective 3 (line 78-79), replace with socio-demigraphic characters;

Response: Corrected. We have edited the whole paper again.

line 257-258 - how do the authors know that consumers 'understand the food 257 supply cost increase and are willing to share that cost'? It was not the scope of the analysis, then it must be proved.

Response: Thanks. This claim is not well supported with evidence in the paper. We have this claim removed.

Reviewer 4 Report

Consumer behaviour is currently a popular topic among researchers. Since the beginning of the pandemic, a large number of articles have been written on the subject in various segments of the economy. The authors unfortunately refer to only 23 items. 
The introduction is very boring and does not reflect the problem studied.
The authors use a lot of acronyms, which are not generally accepted and make the article terribly difficult to read.
WTP, CV, CVM, DBDC - please write these in full names. The full name is written at the beginning of the article, and the reader at the end will already forget what the acronym stood for.
Table 2. you can't describe gender with a 0-1 system. what the authors come out with is that the average gender is 0.56 Female-Male!!!
In which program the statistics were done should also be completed. In Table 3 the authors have included the regression results. Please present the results in the form of a mathematical formula - which will be more readable.
In the introduction, the authors mentioned that the aim of the research includes 3 areas - I do not see this reflected in the results.
No discussion - the authors do not confront their results with the results of other researchers.  There have been many publications on consumer behaviour in food markets at the time of COVID. They are available in various databases e.g. Scholar, MDPI and others.
The summary is not suitable either, it presents loose devotions of the authors not supported by scientific results. The subsections limitations and strengths of the article are missing.
Bibliography is also unacceptable - very poor.

Author Response

We use bold type face to respond to each comment.

Consumer behaviour is currently a popular topic among researchers. Since the beginning of the pandemic, a large number of articles have been written on the subject in various segments of the economy. The authors unfortunately refer to only 23 items. 

ResponseWe have now increased to 49.

The introduction is very boring and does not reflect the problem studied.

Response: We have added qutie a few references, and explitely identified the literature gap and how our paper would fill it.  

The authors use a lot of acronyms, which are not generally accepted and make the article terribly difficult to read.
WTP, CV, CVM, DBDC - please write these in full names. The full name is written at the beginning of the article, and the reader at the end will already forget what the acronym stood for.

Response: We reduced teh number of acronums, spelling our all DBDC, CV and CVM, only keeping WTP as it is used frequently throughout the article.

Table 2. you can't describe gender with a 0-1 system. what the authors come out with is that the average gender is 0.56 Female-Male!!!

Response: We have many dummy variables that take 0 and 1 as values. The mean represents the share of observation 1 among all observations.  This is easily understandable by readers and is commonly used in publications (see #11, #48). Nevertheless, we added a note under the table.

In which program the statistics were done should also be completed.

Response: All analyses are completed in STATA. We added a sentence in the result seciton.

In Table 3 the authors have included the regression results. Please present the results in the form of a mathematical formula - which will be more readable.

Response: The mathematical formula are given in equations (2) and (3). Four model specifications (with different dependent and independent variables) are estimated.  Table is the most efficient way to present them, as is in almost all the publications we cited.  It seems the other three reviewers have no problem with the table presentation.    

In the introduction, the authors mentioned that the aim of the research includes 3 areas - I do not see this reflected in the results.

Responses:  The objective 1) to what extent consumers are willing to pay is addressed in table 4—141% to 200% higher; 2) the role people’s expectation of the epidemic duration on the WTP is addressed in table 3—positvely influencing WTP; and 3) socio-economic characteristics differences reflected on COVID impact on consumption is also addressed in table 3—COVID induced WTP premium is lower for females than males, higher for high income people, etc.

No discussion - the authors do not confront their results with the results of other researchers.  There have been many publications on consumer behaviour in food markets at the time of COVID. They are available in various databases e.g. Scholar, MDPI and others.

Response We have combined the results and discussion section together, as is allowed by the journal format (“ This section may be combined with Results.”).  We have added references about COVID and consumer food preference (#6, #7, #31).

The summary is not suitable either, it presents loose devotions of the authors not supported by scientific results.

Responses: The conclusion is revised. Each paragraph is based on one empirical result, and we derive some implicatoin to government and/or industry policy based on that result.

The subsections limitations and strengths of the article are missing.
Responses: the journal format does not require such as seciton. We have added our limitations in the conclusion section.

Bibliography is also unacceptable - very poor.

Responses: We have more than doubled the references.

Round 2

Reviewer 1 Report

Dear Authors,

thank you for considering my suggestions and I appreciate your response. The presented version of the article is more correct. After reviewing the content of the new Foods - manuscript 1353755 ‘Does the COVID-19 pandemic change consumers’ food consumption and willingness-to-pay? The Case of China' I would like to conclude that the issue addressed is in line with the theme of the journal. The problem presented in the article is important and interesting, the research in this area is needed.

The structure of the publication is appropriate, the goals are clearly defined. The methodology used is good, correctly described and supported by literature. The conclusions of the analysis presented are correct and in line with the stated objectives. More literature has been added (a total of 49 references instead of 23). Relevant literature has been included in the study, although this part could still be supplemented. For example in the lines 45-61 it is worth adding that willingness-to-pay is a bahavioural component of sustainable attitudes. WTP might be used as a segmentation factor of food consumers. Wrote about it, among others
1. Gazdecki, M.; Goryńska-Goldmann, E.; Kiss, M.; Szakály, Z. Segmentation of Food Consumers Based on Their Sustainable Attitude. Energies 2021, 14, 3179.
2. Gerini, F.; Alfnes, F.; Schjøll, A. Organic- and Animal Welfare-labelled Eggs: Competing for the Same Consumers? J. Agric. Econ. 2016, 67, 471–490.
3. Janßen, D.; Langen, N. The bunch of sustainability labels—Do consumers differentiate? J. Clean. Prod. 2017, 143, 1233–1245.
4. Miller, S.; Tait, P.; Saunders, C.; Dalziel, P.; Rutherford, P.; Abell, W. Estimation of consumer willingness-to-pay for social responsibility in fruit and vegetable products: A cross-country comparison using a choice experiment. J. Consum. Behav. 2017, 16, e13–e25.
5. La Lama, G.C.M.-D.; Estévez-Moreno, L.X.; Villarroel, M.; Rayas-Amor, A.A.; María, G.A.; Sepúlveda, W.S. Consumer Attitudes Toward Animal Welfare-Friendly Products and Willingness to Pay: Exploration of Mexican Market Segments. J. Appl. Anim. Welf.Sci. 2018, 22, 13–25.
6. Voon, J.P.; Ngui, K.S.; Agrawal, A. Determinants of willingness to purchase organic food: An exploratory study using structural equation modeling. Int. Food Agribus. Manag. Rev. 2011, 14, 103–120.There are a discussion with references to other studies. Summary: more text has been added.

Based on these comments, I suggest a minor revision of the article.

Good luck with you!

Author Response

Thank you very much for the good points!  We have now further revised this section as you suggested and included these citations.

Reviewer 4 Report

The authors did not follow the comments.
The article needs to be corrected.
I understand very well what regression is and how to read it. Only when publishing an article in a scientific journal for readers of a wide range should the results be presented in such a way that everyone can understand them, not just a statistical specialist.
The authors cite that the discussion is not mandatory and is not required by the journal format. I believe that discussion is an important part of academic work. The place where the obtained results are compared with the results of other specialists. If the authors feel that this section of their article is unnecessary, they may need to search for a popular magazine. Separating the limitations and strengths of the work from the whole text would also help the reader to quickly identify the main values of the work.

Author Response

Only when publishing an article in a scientific journal for readers of a wide range should the results be presented in such a way that everyone can understand them, not just a statistical specialist.

Response: We added the regression model results in mathematical equation format in the Appendix. In this way, readers from different disciplines can read the model results in their preferred formats, i.e., either table format or equation format.

We added a sentence “Regression results in equation format are available in the Appendix.” in the first paragraph of the Empirical Results and Discussion section to point this out to readers.

The authors cite that the discussion is not mandatory and is not required by the journal format. I believe that discussion is an important part of academic work. The place where the obtained results are compared with the results of other specialists. If the authors feel that this section of their article is unnecessary, they may need to search for a popular magazine. Separating the limitations and strengths of the work from the whole text would also help the reader to quickly identify the main values of the work.

Response: We agree with the reviewer that discussion section is a very important component for this paper. In the former round of revisions, we incorporated all the four reviewers’ comments about result discussion which helped improve the discussions substantially. Thank you! Instead of separating the Results section and Discussion section, we combined the two of them. We followed a format that after reporting the results of each significant coefficient, we interpreted the coefficient and thoroughly discussed the reasoning, implications, comparison to other literatures, etc. If we separate the results and discussions, we will have to repeat mentioning the estimated coefficient and interpretation in the Discussion section, which may confuse readers.

To address the reviewer’s comment, we added a sentence “In the following, we report the estimated coefficients that are statistically significant, interpret the results, and discuss the findings.” at the end of the first paragraph of the Empirical Results and Discussion section so that readers can clearly see that our discussions follow the coefficient report and interpretation.